# Position: AI Scaling: From Up to Down and Out

**Yunke Wang** [1] [*]  **Yanxi Li** [1] [*]  **Chang Xu** [1]

## Abstract

AI Scaling has traditionally been synonymous with Scaling Up, which builds larger and more powerful models. However, the growing demand for efficiency, adaptability, and collaboration across diverse applications necessitates a broader perspective. This position paper presents a holistic framework for AI scaling, encompassing Scaling Up, Scaling Down, and Scaling Out. It argues that **while Scaling Up of models faces inherent bottlenecks, the future trajectory of AI scaling lies in Scaling Down and Scaling Out**. These paradigms address critical technical and societal challenges, such as reducing carbon footprint, ensuring equitable access, and enhancing cross-domain collaboration. We explore transformative applications in healthcare, smart manufacturing, and content creation, demonstrating how AI Scaling can enable breakthroughs in efficiency, personalization, and global connectivity. Additionally, we highlight key challenges, including balancing model complexity with interpretability, managing resource constraints, and fostering ethical development. By synthesizing these approaches, we propose a unified roadmap that redefines the future of AI research and application, paving the way for advancements toward Artificial General Intelligence (AGI).

## 1. Introduction

The field of artificial intelligence (AI) has witnessed extraordinary advancements over the past decade, largely driven by the relentless pursuit of Scaling Up. Early breakthroughs were characterized by models with millions of parameters, such as AlexNet (Krizhevsky et al., 2012), word2vec (Church, 2017) and BERT (Devlin et al., 2019),

---
[*]Equal contribution [1]School of Computer Science, The University of Sydney, Sydney, Australia. Correspondence to: Chang Xu <c.xu@sydney.edu.au>.

*Proceedings of the 42nd International Conference on Machine Learning*, Vancouver, Canada. PMLR 267, 2025. Copyright 2025 by the author(s).

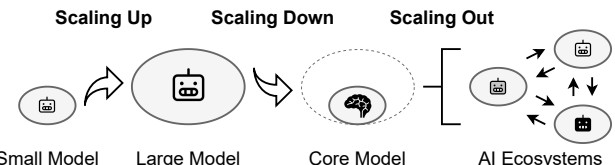

*Figure 1.* The proposed framework for AI Scaling that integrates: (a) **Scale Up** increases model size and complexity, enhancing performance but demanding more computational resources. (b) **Scale Down** reduces model size and distills the essence of these systems into a smaller, more efficient core model. (c) **Scale Out** leverages the core model to derive multiple task-specific interfaces, enabling adaptation to diverse tasks and interaction with the environment.

which paved the way for deep learning's success. This progression quickly escalated to models with billions of parameters, exemplified by GPT-3 (175 billion parameters) (Brown et al., 2020) and more recently GPT-4 (Achiam et al., 2023), which has further expanded the boundaries of language understanding and generation. Similarly, vision-language models like CLIP (Radford et al., 2021) and Flamingo (Alayrac et al., 2022) have showcased the transformative power of scaling multimodal architectures. These advancements highlight how Scaling Up has enabled AI systems to achieve remarkable generalization and versatility across diverse tasks.

However, as Scaling Up progresses, the field faces a critical bottleneck: data (Shumailov et al., 2024). The success of scaling has been largely contingent on the availability of massive, high-quality datasets. Foundational datasets like Common Crawl[1] and large-scale multimodal Corpus have been extensively mined, leaving diminishing returns from further expansion. While multimodal data sources remain an underexplored frontier, their integration presents unique challenges, including alignment across modalities and domain-specific constraints. Moreover, the cost of processing this data at scale, in terms of both computational energy and infrastructure demands, compounds the difficulty of sustaining the current paradigm. These challenges underscore a pivotal question: *can Scaling Up alone continue to deliver transformative progress, or are new paradigms required to achieve the ultimate vision of Artificial General Intelligence (AGI)*?

This position paper presents a holistic framework for AI

---
[1]https://commoncrawl.org/

scaling (Fig. 1), encompassing Scaling Up, Scaling Down, and Scaling Out. It argues that **while Scaling Up of models faces inherent bottlenecks, the future trajectory of AI scaling lies in Scaling Down and Scaling Out**. They form a natural progression in the evolution of AI, each building on the achievements and limitations of the previous. Scaling Up represents the exploratory frontier, pushing the boundaries of model performance by increasing parameter counts, training on vast datasets, and leveraging unprecedented computational resources. This phase is critical, as it establishes benchmarks and reveals the upper limits of what AI systems can achieve. For instance, Scaling Up to models like GPT-4 demonstrates the potential for generalization across tasks, offering a roadmap for what optimal performance could look like across diverse domains.

However, Scaling Up comes at a cost—computational, financial, and environmental. These costs, coupled with the diminishing returns due to data saturation, necessitate a shift to Scaling Down. Importantly, Scaling Down is guided by the insights gained from Scaling Up. By analyzing the structure and performance of large-scale models, researchers can identify redundancies, prune parameters, and distill the essence of these systems into smaller, more efficient models. Scaling Down thus becomes a process of optimization, reducing model size while retaining or even enhancing performance for specific tasks. For example, foundational models with hundreds of billions of parameters can be scaled down to a fraction of their size, enabling deployment on edge devices and in resource-constrained environments. This phase democratizes AI, making advanced capabilities accessible to a broader audience and reducing barriers to entry for smaller organizations.

Scaling Out builds upon the advancements achieved through Scaling Down, leveraging lightweight and efficient models to enable large-scale deployment across distributed environments. Rather than relying solely on a single, monolithic AI model, Scaling Out envisions an AI ecosystem where a foundation model serves as the core intelligence, from which specialized models emerge to address specific tasks or domains. These specialized models interact with the world through *structured interfaces*, such as APIs or intelligent agents, forming a decentralized network of AI-driven applications. For example, on content creation platforms like TikTok, YouTube, and Instagram, foundation models such as LLaMA (Touvron et al., 2023) and QWEN (Yang et al., 2024) have given rise to numerous fine-tuned variants that cater to distinct creative needs, such as AI models optimized for video scriptwriting, personalized recommendation, or style-consistent image generation. These models, acting as interfaces, empower both human creators and AI-driven content generation pipelines, fostering a dynamic ecosystem where AI seamlessly integrates into workflows. Scaling Out does not seek to replace human creativity but instead enhances global cultural exchange by providing adaptable, interactive AI systems that extend AI's reach beyond isolated models into an interconnected and ever-evolving ecosystem.

The progression from Scaling Up to Scaling Down and then Scaling Out is not merely sequential but interdependent. Scaling Up defines the theoretical and practical benchmarks for AI performance. Scaling Down operationalizes these benchmarks, ensuring they are achievable in diverse environments. Scaling Out amplifies these capabilities, enabling AI to thrive in dynamic, decentralized systems. Together, these paradigms form a cohesive framework that transforms AI from a centralized, high-resource endeavor into a distributed, inclusive, and adaptive force capable of addressing humanity's most complex challenges.

Notably, while our primary discussions in this paper center around LLMs, the proposed framework of Scaling Down and Scaling Out is not limited to them. The techniques and principles involved such as model compression, pruning and decentralized coordination are widely applicable across various AI systems, including vision models, reinforcement learning and multimodal architectures. By using LLMs as illustrative cases, we aim to ground the framework in a currently prominent domain, while maintaining its broader relevance.

## 2. Scaling Up: Expanding Foundation Models

Scaling Up is critical for advancing AI research and applications as it pushes the boundaries of what AI systems can achieve. Larger models act as high-quality foundational models for both academia and industry, further setting benchmarks and inspiring further innovations. These models are capable of solving a wide range of tasks while they can serve as a foundation for creating specialized and diverse AI interfaces through fine-tuning as well.

### 2.1. Scaling in AI models

The past experience in AI Scaling Up is mostly based on increasing data size, model size and computational resources.

**Data Size.** Expanding dataset size is a fundamental aspect of Scaling Up AI models, as it directly impacts the quality of the system. Large and diverse datasets expose models to a wide variety of knowledge, thereby enabling them to perform effectively across multiple domains. For instance, GPT-3 (Brown et al., 2020) was trained on 570GB of cleaned and curated text data drawn from sources such as Common Crawl, BooksCorpus, and Wikipedia, which enabled it to generate human-like responses across diverse contexts. More recently, multi-modal datasets such as LAION-5B (Schuhmann et al., 2022) have been used to scale vision-language models like Stable Diffusion (Rombach et al., 2022), showcasing the impact of data size on

model's capabilities.

**Model Size.** Larger models have greater representational power, allowing them to capture complex relationships within data. For example, the 175B parameters of GPT-3 significantly outperform their predecessors in tasks that require learning of few shots or zero shots (Brown et al., 2020). Similarly, GLaM (Du et al., 2022) scaled to 1.2 trillion parameters using a mixture of experts, activating only a subset of parameters per task, which reduced computational costs while maintaining high performance. The scaling laws proposed by (Kaplan et al., 2020) highlight that model performance improves predictably with increased size. This insight has guided the development of increasingly large models, unlocking capabilities like in-context learning and cross-modal understanding.

**Computational Resources.** The process of scaling computational resources has evolved dramatically alongside advancements in AI, particularly in computer vision and NLP. Early in the development of these fields, training models required only a few plain GPUs. For instance, AlexNet (Krizhevsky et al., 2012), which revolutionized computer vision in 2012, was trained using just two GTX 580 GPUs. Similarly, early NLP models such as Word2Vec (Church, 2017) were trained on modest computational setups. However, the era of LLMs has accelerated in unprecedented demands for computational resources. For example, OpenAI's GPT-3 (Brown et al., 2020) has 175B parameters and requires 10,000 NVIDIA V100 GPUs for training, consuming an estimated 1,287 MWh of electricity. Meanwhile, Meta's LLaMA 3 (Touvron et al., 2023) scaled training to utilize thousands of NVIDIA A100 GPUs, representing the latest generation of high-performance accelerators optimized for AI workloads. This progression highlights the critical role of computational scaling in AI model's performance.

## 2.2. Bottleneck

From the *data* perspective, as pointed out by many researchers, large-scale pretraining has already utilized most of the high-quality publicly available data on the web. The remaining data is either low-quality or consists of AI-generated content, which risks model degradation due to data contamination and reinforcement of biases (Shumailov et al., 2024). Simply increasing the dataset size will no longer yield the same level of improvement as before. From the *model* perspective, while increasing parameters has led to substantial performance gains in recent years, the returns on scaling have shown diminishing improvements, and larger models suffer from inefficiencies such as redundancy in representation, overfitting to training distributions, and difficulties in interpretability and controllability. Additionally, the training and inference of massive models introduce challenges in optimization stability and robustness (Dai et al., 2024b). From the *computational resource* aspect, the exponential growth in required hardware, energy consumption, and costs is reaching unsustainable levels. The marginal benefit of adding more compute is decreasing while the environmental impact is rising (Wu et al., 2024). The availability of high-performance GPUs poses financial constraints that limit the feasibility of further scaling. Together, these bottlenecks indicate that the traditional approach of scaling up is approaching its practical limits.

## 2.3. Future Trends

Despite bottlenecks in AI scaling, Scaling Up remains essential for pushing AI model's performance boundary. The future of Scaling Up should lie in balancing efficiency, adaptability and sustainability to meet the demands of larger models. Innovations in dataset optimization, efficient training, and test-time scaling will redefine AI Scaling Up.

**Dataset Optimization.** As AI continues to scale, data optimization will become a cornerstone for advancing model efficiency and robustness. Future trends will focus on data-efficient training using smaller, high-quality datasets for faster learning. Curriculum learning (Bengio et al., 2009) and active learning (Settles, 2009) will help models acquire knowledge incrementally and prioritize impactful samples. Techniques for handling noisy data, such as noise-robust loss functions and data augmentation, will enhance model resilience. Additionally, leveraging proprietary, domain-specific datasets will drive breakthroughs by providing richer insights beyond public data.

**Efficient Training.** Another trend is developing efficient training methods to address the growing computational and environmental costs of training large models. Progressive training, where models gradually scale from smaller submodels to full-capacity systems, will become a standard approach to reduce resource demands in the initial stages. Distributed optimization techniques, such as asynchronous training paradigms, will improve scalability across large computational infrastructures. Advances in mixed-precision training, sparse updates, and activation checkpointing will further minimize memory and compute overhead, making AI development more sustainable and scalable.

**Test-time Scaling.** Recent research has highlighted the potential of scaling up test-time computing to enhance the performance of large language models (LLMs), providing an alternative to solely scaling up model parameters. For example, Snell et al. (2024) explore two strategies: adaptive output distribution and verifier-based search mechanisms, both improving model performance dynamically. Unlike previous inference-time optimization attempts, this approach tailors compute allocation to problem complexity, enabling smaller models to outperform larger ones on certain prompts.

Adaptive test-time scaling presents a promising direction for optimizing efficiency without excessive pretraining.

## 3. Scaling Down: Refining Core Functions

As models become increasingly large and complex through *Scaling Up*, their training, deployment, and maintenance demand significant computational, memory, and energy resources. These challenges limit accessibility and scalability. A critical question emerges: *how can we maintain or improve model effectiveness while reducing size and computational requirements?* Drawing inspiration from the human brain, where specialized small units handle essential functions while auxiliary components support adaptability and memory, the **Scaling Down** concept offers a novel approach. By identifying and extracting the essential functional modules of large models, Scaling Down makes it possible to reduce the model size and computation costs significantly while retaining or even enhancing key capabilities. Scaling Down can be approached in *two distinct ways*. The first involves directly reducing the model size by decreasing the number or precision of parameters (Section 3.1). Alternatively, minimizing redundant or unnecessary computations can enhance computational efficiency without altering the number or precision of parameters (Section 3.3).

### 3.1. Reducing the Size of Large Models

The most straightforward approach to reducing model size involves reducing the number of parameters within a model. **Pruning** achieves this by simplifying neural networks through the removal of less significant components (Le-Cun et al., 1989; Han et al., 2015; Molchanov et al., 2016). LLM-Pruner (Ma et al., 2023) proposes a task-agnostic approach to structural pruning by selectively removing non-critical structures using gradient information. Wanda (Sun et al., 2024) emphasizes simplicity and efficiency by pruning weights based on the product of weight magnitudes and corresponding input activations without the need for retraining or weight updates.

An alternative to directly removing parameters is the use of **low-rank approximations**, which employ smaller matrices to approximate larger ones Sainath et al. (2013). Low-Rank Adaptation (LoRA) (Hu et al., 2021) tackles the inefficiency of fine-tuning all model parameters by introducing trainable low-rank decomposition matrices into Transformer layers while keeping the pre-trained model weights frozen. Linformer (Wang et al., 2020) leverages the observation that self-attention mechanisms in Transformers exhibit low-rank structures. By approximating the self-attention matrix with a low-rank factorization, Linformer reduces the time and space complexity of self-attention to a linear scale.

Another effective strategy focuses on reducing parameter precision rather than quantity. **Quantization** reduces the bit-width of weights and activations by substituting floating-point parameters with integers (Gupta et al., 2015; Nagel et al., 2020). GPTQ (Frantar et al., 2022) introduces an efficient one-shot weight quantization method based on approximate second-order information. AWQ (Lin et al., 2024) focuses on activation-aware weight quantization, leveraging the unequal importance of weights and optimal per-channel scaling to protect salient weights. QLoRA (Dettmers et al., 2023) introduces a memory-efficient fine-tuning approach by combining 4-bit quantization with LoRA.

Rather than modifying existing models, **knowledge distillation** (KD) facilitates the transfer of knowledge from large and complex teachers to small and efficient students (Hinton, 2015). The students are trained to replicate the behavior of the teachers. Yu et al. (2024) propose to distill System 2 reasoning processes, such as Chain-of-Thought and System 2 Attention, into a single-step System 1 model, eliminating intermediate reasoning while retaining or improving task performance. Program-aided Distillation (PaD) (Zhu et al., 2024) introduces a new KD paradigm that uses reasoning programs to verify and refine synthetic CoT data, enhancing distilled reasoning quality. PaD automates error-checking, incorporates iterative self-refinement to address faulty reasoning chains, and employs step-wise beam search to validate reasoning steps progressively.

### 3.2. Reducing the Scale of Training

Recent efforts in dataset pruning have aimed to reduce training costs while preserving model performance, particularly on large-scale datasets. Traditional snapshot-based pruning methods estimate sample importance using static predictions from early or late training epochs but fail to generalize across architectures. To address these issues, He et al. (2023) propose Dynamic Uncertainty, which captures prediction variance over time using sliding windows, enabling better distinction between hard samples. Building on the need to incorporate temporal dynamics, the Temporal Dual-Depth Scoring (TDDS) (Zhang et al., 2024) method introduces a two-tiered scoring system that tracks both the stability and contribution of samples throughout training.

In the multimodal setting, Sieve (Mahmoud et al., 2024) addresses noise in web-crawled image-text pairs by using image captioning models and sentence transformers. YOCO (He et al., 2023) bridges the gap between dataset condensation and pruning by introducing Logit-Based Prediction Error and Balanced Construction that enable flexible resizing of condensed datasets without repeating the condensation process, offering a practical solution for constrained training environments. Together, these works reflect a shift toward dynamic, scalable, and generalizable data selection methods across both unimodal and multimodal domains.

### 3.3. Optimizing Computational Efficiency

**Speculative decoding** optimizes the inference process by dynamically adapting decoding strategies. Leviathan et al. (2023) introduce speculative decoding as a method that leverages more efficient approximation models to propose candidate tokens, which are then verified by the target model in parallel. Similarly, Chen et al. (2023) propose speculative sampling, employing a draft model to generate multiple token candidates, which are then validated using a modified rejection sampling scheme. These methods underscore the potential of speculative execution to mitigate the inherent inefficiencies of autoregressive decoding, enabling faster inference without retraining or compromising output quality.

**Key-value cache** is a pivotal strategy in autoregressive decoding, where intermediate states of attention mechanisms are stored to avoid recomputation in subsequent inference steps. This technique significantly accelerates the generation of long sequences by leveraging stored key-value pairs from previous layers. However, it introduces additional memory overhead, which must be carefully managed. Sparse attention mechanisms (Zhang et al., 2023c; Anagnostidis et al., 2024; Liu et al., 2024c) use specialized sparsity patterns that prevent unnecessary token access. They use KV cache eviction and compression strategies to achieve significant improvements in latency, throughput, and memory savings. Block-wise KV cache management (Kwon et al., 2023; Prabhu et al., 2024) adopts memory fragmentation techniques inspired by paged memory systems, offering efficient runtime memory allocation and reallocation.

**Mixture of Experts** (MoE) introduced distributed specialization, enabling efficient scaling through task-specific submodels controlled by a gating mechanism (Jacobs et al., 1991). Early dense MoE models suffered computational inefficiencies (Jordan & Jacobs, 1994). Sparse architectures (Shazeer et al., 2017) improved efficiency by selectively activating relevant experts. Models like GShard (Lepikhin et al., 2020), Switch Transformer (Fedus et al., 2022), and GLaM (Du et al., 2022) leveraged MoE for state-of-the-art performance with reduced computation. Recent advances, including Mixtral (Jiang et al., 2024) and DeepSeekMoE (Dai et al., 2024a), further optimized efficiency.

### 3.4. Small Models for Large Impacts

Designing high-efficiency architectures is fundamental to developing small-scale models. The most computationally intensive and memory-intensive component of Transformer-based models is the Attention mechanism. Extensive research efforts have been devoted to enhancing the efficiency of Attention mechanisms. Notable advancements include *Flash Attention* (Dao et al., 2022), which is utilized by models such as Phi-1.5 (Li et al., 2023) and DeepSeek-LLM (Bi et al., 2024), *Grouped Query Attention* (Ainslie et al., 2023),

which is utilized by MiniCPM (Hu et al., 2024), Mistral (Jiang et al., 2023), Phi-3 (Abdin et al., 2024), DeepSeek-LLM (Bi et al., 2024), and DeepSeek-V2 (Liu et al., 2024a), and *Multi-Head Latent Attention*, which was first introduced by DeepSeek-V2 (Liu et al., 2024a) and has been adopted in its successor, DeepSeek-V3 (Liu et al., 2024b).

While such innovations enable the development of highly efficient small models, further improvements are necessary to bridge the performance gap between small and large-scale models. Key directions for achieving this include curating high-quality training data, designing scalable training strategies, and leveraging techniques such as mixture-of-experts (MoE), which allow for the selective activation of model components to optimize efficiency and performance.

**High-Quality Training Data.** The Phi family of models (Gunasekar et al., 2023; Li et al., 2023; Javaheripi et al., 2023; Abdin et al., 2024) highlights the importance of high-quality training data. Rather than relying on vast amounts of noisy web-scraped text, these models are trained on curated, synthetically generated textbook-style data, including structured exercises and carefully filtered educational content. This approach enhances efficiency and mitigates common issues such as hallucination and bias.

**Scalable Training Strategies.** Training efficiency is another critical factor in developing compact yet powerful models. Mini-CPM (Hu et al., 2024) introduces Model Wind Tunnel Experiments (MWTE) to optimize hyperparameter selection, ensuring that smaller models are trained in a computationally efficient manner. Additionally, it employs the Warmup-Stable-Decay (WSD) learning rate scheduler, which segments training into distinct phases to maximize hardware utilization and improve convergence.

**More Parameters but Less Activation.** A crucial trend in optimizing smaller models for efficiency is the adoption of MoE, where a subset of model parameters is activated per token, reducing computation while maintaining a large overall parameter pool. Several recent models exemplify this technique: Mixtral (Jiang et al., 2024) consists of 8 expert models with a total of 56B parameters, while each token is processed by only 2 experts. Phi-3.5-MoE (Abdin et al., 2024) comprises 16 experts totaling 60.8B parameters but activates only 6.6B (10.9%). DeepSeek-V2 (Liu et al., 2024a) is a 236B-parameter model with 21B parameters activated per token (8.9%). DeepSeek-V3 (Liu et al., 2024b) scales further to 671B total parameters while activating only 37B per token (5.5%).

### 3.5. Future Trends

**Core Functional Module Refinement.** A promising direction for future research in Scaling Down models lies in refining core functional modules. While existing methods

predominantly emphasize the balance between efficiency and effectiveness, a critical gap remains in identifying the minimal functional module within large models. This minimal module would represent the smallest possible unit that retains all essential functionalities without compromising performance. Future investigations may focus on developing systematic approaches to detect and characterize such modules, potentially leveraging advancements in model pruning and knowledge distillation. Establishing rigorous criteria for defining and verifying minimal functional modules could significantly contribute to optimizing model architectures while maintaining their operational integrity.

**External Assistance.** Leveraging external assistance enables small-scale core models to dynamically extend their capacity to handle complex tasks. Retrieval-Augmented Generation (RAG) (Lewis et al., 2020) is a method for *external knowledge* augmentation. RAG combines pre-trained parametric memory with non-parametric memory, which enables models to fetch contextually relevant information dynamically. Integrating *external tools* allows models to assign specialized operations to certain systems. Toolformer (Schick et al., 2023) can autonomously learn to invoke external APIs, such as calculators, search engines, and translation systems. Beyond merely utilizing external tools, recent advancements suggest that models can also generate tools to extend their own capabilities. VISPROG (Gupta & Kembhavi, 2023) can leverage in-context learning to produce modular, Python-like programs and execute them for complex visual reasoning tasks.

# 4. Scaling Out: Advancing AI Ecosystems

Scaling Up and Scaling Down represent two complementary approaches to AI scaling, yet neither fully realizes AI's potential in real-world applications. Scaling Up builds larger, generalized models like GPT and BERT, but their resource demands limit accessibility and task-specific adaptability. Scaling Down optimizes models for efficiency, enabling deployment in resource-constrained environments, but struggles with adaptability, collaboration, and decentralized intelligence. To address these gaps, AI must evolve into a distributed ecosystem where multiple AI entities interact, specialize, and collectively enhance intelligence.

We propose Scaling Out as the next step in AI evolution, which is a paradigm that leverages efficient, task-specific AI models derived from large-scale foundation models, distributed across networks and interfacing through modular, interactive systems. Scaling Out can expand AI's reach by deploying interfaces, which enable AI to interact with users, devices, and other systems. *These **interfaces**, powered by specialized sub-models derived from foundation models, form an expandable AI ecosystem.* Unlike Scaling Up's focus on size or Scaling Down's focus on efficiency, Scaling

Out emphasizes accessibility and adaptability. For example, in a smart city, AI interfaces for traffic, energy, and safety could collaborate to create a seamless urban experience, showcasing Scaling Out's transformative potential.

## 4.1. Scaling Out builds an AI Ecosystem

Scaling Out transforms isolated AI models into a diverse, interconnected ecosystem by expanding foundation models like LLaMA (Touvron et al., 2023) and Stable Diffusion (Rombach et al., 2022) into specialized variants equipped with structured interfaces. Foundation models provide generalized intelligence, while specialized models, fine-tuned for tasks like legal contract analysis or medical diagnosis, ensure domain-specific adaptability. For instance, ControlNet (Zhang et al., 2023b) enables structured image generation by conditioning outputs on additional inputs, demonstrating how foundation models can be adapted for specific use cases.

Interfaces bridge specialized models with users, applications, and other AI systems. These range from simple APIs for task-specific queries to intelligent agents capable of multi-turn reasoning and decision-making. For example, the GPT Store hosts specialized GPTs, which are sub-models derived from the GPT Foundation Model that perform tasks like coding assistance and creative writing. Similarly, Hugging Face's ecosystem fine-tunes LLaMA variants for tasks such as sentiment analysis and summarization, showcasing how Scaling Out extends AI's reach across domains.

*By combining foundation models, specialized variants, and well-designed interfaces, Scaling Out creates a dynamic AI ecosystem.* This ecosystem fosters collaboration, enables large-scale deployment, and continuously expands AI's capabilities, marking a shift toward open, scalable, and domain-adaptive AI infrastructure.

To make this concept tangible, we provide some analogies and real-world illustrations. (i) In a **functioning society**, individuals often specialize in different roles like teachers, policemen, and engineers. Each adapts to their context while contributing to maintaining social order and driving progress. For example, a teacher adjusts their methods to meet the needs of students, while contributing to the broader educational infrastructure. This mirrors how Scaling Out in AI enables diverse, task-specific models to operate within their domains, while remaining connected through structured interfaces. (ii) In **healthcare systems**, when diagnosing and treating a patient, hospitals do not rely on a single generalist. Instead, responsibilities are distributed across specialized professionals such as GPs handling basic diagnostics, radiologists interpreting medical images, surgeons performing targeted procedures and pharmacists managing prescriptions. Each operates based on their expertise and the specific contextual information they receive. These ex-

perts interact through structured protocols (*e.g.*, referrals), and their collaboration results in an adaptive, efficient, and context-aware response tailored to individual needs. This reflects how AI systems under Scaling Out can coordinate modular intelligence across specialized sub-models.

## 4.2. Technical Foundations

Scaling Out relies on efficiently adapting foundation models into specialized models for different tasks and domains. Traditional fine-tuning requires extensive computational resources, but **Parameter-Efficient Fine-Tuning** (PEFT) techniques allow models to be adapted efficiently while preserving the original knowledge. Methods like LoRA (Hu et al., 2021) and Adapter Layers enable adding task-specific knowledge without modifying the entire model. Prompt Tuning and Prefix Tuning (Li & Liang, 2021) further optimize the behavior of the model by modifying inputs rather than parameters. These techniques are widely used in HuggingFace's Transformers and applications like BloomZ, which enables multilingual fine-tuning of large models with minimal computational cost (Muennighoff et al., 2022).

**Condition control** enables a single foundation model to dynamically adapt to multiple tasks without the need for retraining distinct models. Instead of fine-tuning a model separately for every task, condition control allows AI models to modify their behavior through additional input constraints, making them more flexible. ControlNet (Zhang et al., 2023a) extends Stable Diffusion by incorporating structural guidance (*e.g.*, edge maps, depth maps and pose estimation) to generate context-aware images while maintaining the efficiency of the original model. Similarly, in large language models, FLAN-T5 (Chung et al., 2024) demonstrates how conditioning input prompts can alter model outputs for diverse tasks like summarization, translation, and reasoning without fine-tuning. In speech synthesis, VALL-E (Wang et al., 2023) utilizes audio conditions to generate highly expressive speech from a short sample, enabling personalized voice generation without retraining on new data.

**Federated learning** (FL) enables the collaborative training of AI models across distributed devices or systems without centralizing data. This decentralized approach ensures data privacy and security, as raw data remains on local devices while only model updates (*e.g.*, gradients) are shared. FL allows specialized sub-models to be trained on diverse, domain-specific datasets, enhancing their adaptability to local conditions and tasks. For example, in healthcare, FL enables hospitals to collaboratively train diagnostic models without sharing sensitive patient data, ensuring compliance with privacy regulations (Yang et al., 2019). Techniques like Federated Averaging (McMahan et al., 2017) optimize communication efficiency, making FL scalable across millions of devices. Additionally, advancements such as Federated

Transfer Learning (Saha & Ahmad, 2021) and Personalized Federated Learning (Smith et al., 2017) further enhance the adaptability of models to heterogeneous data distributions, a key requirement for Scaling Out.

**Protocol and System-Level Coordination.** As Scaling Out increasingly depends on collaboration among specialized models and agents, standard communication protocols play a key role. Google's recent *Agent-to-Agent (A2A)*[2] initiative introduces an open protocol that enables agents built on different frameworks to interoperate, negotiate interaction formats, and securely collaborate. Similarly, the emerging *Model Context Protocol (MCP)*[3] provides a standardized interface for model-to-model interactions. MCP facilitates consistent context sharing, input/output formatting, and execution state management across diverse models.

## 4.3. Future Trends

**Blockchain.** Just as App stores in Android/iOS provide diverse applications, an AI model store will emerge, enabling users to access, customize, and deploy specialized AI models. For example, the recently launched foundation model DeepSeek-v3 (Liu et al., 2024b) has already surpassed 100 variations in just one month, demonstrating how foundational models can rapidly evolve into specialized versions. To ensure security, transparency, and intellectual property protection in decentralized AI marketplaces, blockchain can serve as a trust layer, recording all modifications, ownership changes, and interactions on an immutable ledger. Every fine-tuning adjustment, API call, or derivative model creation would leave a verifiable trace, ensuring credit attribution, preventing unauthorized modifications, and securing proprietary AI advancements. This decentralized framework will safeguard AI innovations and ensures a collaborative, accountable AI ecosystem, where Scaling Out thrives on trustworthy, trackable, and openly governed AI interfaces.

**Edge Computing.** Edge computing processes data locally on devices, such as smartphones, IoT sensors, or edge servers, minimizing the need to send information to centralized data centers. Federated learning complements this by allowing distributed devices to collaboratively train models without sharing raw data. Together, these technologies reduce latency, improve real-time decision-making, and ensure scalability by distributing computation across a network of edge nodes. For Scaling Out, this decentralized architecture allows billions of lightweight, specialized AI agents to operate independently while sharing collective insights, as seen in applications like personalized healthcare monitoring or real-time traffic management. This synergy fosters ecosystems where agents adapt locally while contributing to a globally optimized intelligence network.

---

[2]https://github.com/google/A2A
[3]https://modelcontextprotocol.io/introduction

# 5. Future Application Prospects

The true potential of AI Scaling lies in the future scenarios it can enable. This section explores two use cases that illustrate the transformative capabilities of AI scaling: human-AI creative communities and smart manufacturing ecosystems.

## 5.1. Human-AI Creative Communities

Content creation platforms like TikTok, YouTube, and Instagram showcase how AI scaling transforms creativity and engagement. Scaling Up integrates vast multimodal datasets, enabling foundation models to analyze trends, predict preferences, and optimize recommendations on a global scale. These models, trained on billions of interactions, continuously evolve to match audience demands. Scaling Down brings AI closer to users, with lightweight models enabling real-time video, music, and AR generation on personal devices. On-device AI also enhances content moderation, ensuring platform safety without heavy computational costs. Scaling Out redefines these platforms as AI-driven ecosystems where specialized AI agents actively participate alongside human users. These AI contributors focus on education, sports, music, and niche domains, generating and engaging with content just as human creators do. For example, an education AI produces real-time tutorials, while a sports AI provides live commentary. AI bots collaborate, such as a music AI partnering with a graphic-design AI to create immersive audiovisual content.

At scale, these platforms evolve into hybrid ecosystems where human and AI creators collaborate seamlessly. The interaction between human and AI creators fosters a dynamic, participatory environment where creativity flourishes without boundaries. As AI bots continuously adapt to cultural shifts and audience feedback, they contribute to a globally inclusive and interactive digital space. Such platforms no longer merely host content but become thriving communities of hybrid human-AI interaction, where collaboration and innovation redefine the boundaries of creativity.

## 5.2. Smart Manufacturing Ecosystems

Manufacturing ecosystems differ from traditional multi-agent systems due to their open, dynamic nature and massive scale, involving suppliers, manufacturers, and distributors as autonomous AI interfaces adapting to constant change. Scaling Up builds foundational models that integrate vast, heterogeneous datasets across sourcing, logistics, production, and consumer behavior, equipping agents with advanced predictive capabilities. Scaling Down tailors these global models into lightweight, task-specific AI, optimizing factory operations, equipment monitoring, and localized supply chain decisions. Scaling Out expands the ecosystem's reach, enabling thousands of AI interfaces to collaborate and compete, such as supplier interfaces negotiating contracts or distributor interfaces optimizing delivery schedules. The synergy between these scaling paradigms creates a self-optimizing, adaptive network, where AI continuously integrates new entrants, eliminates inefficiencies, and responds dynamically to global challenges. It transforms manufacturing into an intelligent, resilient ecosystem.

# 6. Challenges and Opportunities

Scaling Up, Down, and Out collectively offers both significant opportunities and notable challenges on the path toward AGI. This section explores these dual aspects, outlining key areas where transformative advancements can occur while addressing critical hurdles that must be overcome.

**Cross-disciplinary research and collaboration.** AI scaling demands cross-disciplinary collaboration. Cognitive science can inspire efficient model architectures, such as modular designs that selectively activate components based on input complexity (Laird et al., 2017). Integrating neuroscience, hardware engineering, and data science is key to achieving adaptive computation at scale. Advancements in hardware efficiency must align with AI scaling. Energy-efficient processors tailored for AI can reduce carbon footprints, while co-developing sparse computation chips enhances Scaling Down, enabling AI in resource-limited settings (James, 2022). Data science defines metrics for AI scaling, establishing benchmarks that balance model size, computational cost, and real-world performance (Kaplan et al., 2020). Standardizing these trade-offs provides a shared framework for innovation, guiding future research and deployment.

**Quantitative metrics and standards for scaling.** Effectively scaling AI requires quantitative models to predict performance and resource trade-offs. Developing scaling metrics for Scaling Down and Scaling Out can help assess efficiency, such as measuring performance gains relative to changes in model size, data, or compute (Kaplan et al., 2020). Formalized metrics also address industry concerns by providing predictable cost-benefit analyses. Scaling laws can estimate energy savings from replacing large models with smaller, task-specific AI, encouraging broader adoption of Scaling Down. Additionally, open benchmarks for Scaling Out should evaluate how distributed models communicate, adapt, and collaborate in real-world tasks, ensuring AI ecosystems remain robust and efficient (Dou et al., 2023).

**Building open ecosystems for lightweight AI.** Scaling Down fosters open and accessible AI ecosystems by enabling lightweight core models as flexible building blocks for diverse applications. Open-source initiatives supported by research and industry can accelerate innovation in this space (Wang et al., 2021). Releasing modular AI components with flexible APIs allows developers to adapt models

for specific needs, such as edge AI in healthcare or resource-efficient industrial applications. These ecosystems also encourage hybrid scaling strategies, combining pre-trained models with task-specific fine-tuning. Industry partnerships are essential for real-world impact. Sectors like agriculture and logistics can benefit from domain-specific AI, and fostering cross-industry collaboration will drive adoption and scalable innovation (Schmidt & Rosenberg, 2014).

**Scaling for sustainability and global equity.** As AI systems expand, their environmental impact grows, making Scaling Down crucial for sustainability. Smaller models can match larger models' performance while consuming less energy (Schwartz et al., 2020). Deploying lightweight AI on solar-powered edge devices reduces reliance on energy-intensive data centers, especially in infrastructure-limited regions. Beyond sustainability, Scaling Out improves AI accessibility, enabling distributed intelligence to serve education, healthcare, and agriculture in underserved areas. For example, offline AI models can assist smallholder farmers with crop management or provide diagnostic tools in rural clinics (Pal & Bi, 2021). Achieving this vision requires aligning AI scaling with societal goals. Governments and organizations should fund scalable AI research that prioritizes sustainability and equity, ensuring AI benefits are broadly and fairly distributed.

**A unified vision toward AGI.** The convergence of Scaling Up, Scaling Down, and Scaling Out forms a cohesive path toward AGI, balancing generalization, efficiency, and adaptability (Bostrom, 2014). Scaling Up builds foundational knowledge, Scaling Down optimizes AGI for diverse environments, and Scaling Out fosters collaboration among specialized intelligence to tackle complex, multidisciplinary challenges (Goertzel & Pennachin, 2006). Achieving this vision requires addressing technical, ethical, and societal challenges. Scaling Up must ensure interpretability and robustness, Scaling Down must prioritize privacy and security, and Scaling Out must foster fairness and accountability in collaborative AI. Cross-disciplinary efforts drawn from cognitive science, hardware engineering, and policy frameworks are essential for sustainable and ethical AGI.

## 7. Alternative Views

While this position paper argues that Scaling Up encounters significant bottlenecks and that future trends will shift towards Scaling Down and Scaling Out, an alternative view is that *Scaling Up remains a viable trajectory despite challenges*. Supporters argue that addressing challenges of Scaling Up is necessary and feasible through interdisciplinary innovation. Key challenges must be overcome include data quality, computational demands, and energy consumption.

Firstly, although data quantity has grown, their quality has not kept pace. Synthetic data offers a controlled alternative but may introduce biases and lack real-world applicability. As for computational demands, computational power is a bottleneck, as traditional hardware faces physical and economic limits. Alternatives like quantum, optical, and neuromorphic computing might help. Concern about energy consumption emphasizes the need for sustainable AI. Low-power chips and integration of renewable energy could reduce the environmental impact of large-scale computing.

In addition, the advantage of increasing the scale of AI models also lies in that it can achieve significant advancements in their reasoning and generalization capabilities. Furthermore, Scaling Up has enabled transformative progress in multi-modal generative AI systems, enhancing the quality and fidelity of generated images and videos which smaller models often struggle to match.

Unlike Scaling Down and Scaling Out, which provide immediate solutions, Scaling Up requires extensive interdisciplinary collaboration. This challenge extends beyond AI research to fields such as hardware engineering, quantum mechanics, and sustainable energy solutions. The long research and development cycles make Scaling Up a long-term strategy. A concern regarding it is technological breakthroughs are unpredictable. Therefore, a balanced strategy is necessary, where both short- and long-term solutions are invested, rather than exclusively focusing on one of them.

## 8. Conclusion

In this paper, we propose a framework for AI scaling, which is from Scaling Up to Scaling Down, then Scaling Out. Scaling Up lays the groundwork with foundation models that generalize across tasks. Scaling Down ensures efficiency and accessibility, optimizing AI for diverse environments. Finally, Scaling Out provides multiple AI interfaces, which enables collaborative intelligence and interaction with users to tackle real-world challenges. Together, these advancements offer a vision of AI that amplifies human creativity, bridges societal divides, and empowers humanity to confront its most ambitious goals. The future of AI is not solely about technological breakthroughs, it is about building systems that are equitable, sustainable, and deeply integrated into the fabric of human progress.

Evaluating Scaling Down and Out requires going beyond accuracy to include efficiency metrics like FLOPs, latency, and energy use. For Scaling Down, metrics such as cost-per-inference, performance-per-watt, and capability-per-dollar better reflect practical value. For Scaling Out, evaluation should consider ecosystem-level indicators such as the scalability of specialized models, robustness of distributed deployments, and diversity of fine-tuned models on open platforms such as Hugging Face or OpenAI's GPT store.

## Acknowledgements

This work was supported in part by the Australian Research Council under Projects DP240101848 and FT230100549.

## Impact Statement

This paper advances AI Scaling by integrating Scaling Up, Scaling Down, and Scaling Out to build efficient, adaptive, and decentralized AI ecosystems. While Scaling Out democratizes AI access across domains like education and healthcare, it also raises concerns about privacy, security, and fairness in decentralized AI marketplaces.

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
