# OpenReview forum: "Position: AI Scaling: From Up to Down and Out"
_ICML.cc/2025/Position_Paper_Track — ICML 2025 Position Paper Track poster_

### Official Review · Reviewer_hbVy · 2025-02-23

**Significance:** 2
**Argument Clarity:** 3
**Rating:** 4
**Confidence:** 4

**Questions:**

- What do authors think about architectural innovations and breakthroughs that can potentially increase the reasoning power of large language models? Before the introduction of attention and transformers, the field of deep learning was also plateauing and one could have argued that it's about time to focus on scaling down and out. While scaling down and out should happen naturally at every phase of innovation in the AI field, it should not be thought as a paradigm shift where training more complex models with more parameters or more advanced architecture is expected to show no additional gain and researchers should shift their attention to scaling down and out. It would be great to hear authors' thoughts on this.

**Discussion Potential:**

3

**Paper Summary:**

The paper suggests that instead of scaling up, AI systems should scale down and out. It reviews scaling up efforts and shares viewpoints about quality publicly available data for training being almost fully explored and the diminishing return over increasing large language model sizes and their carbon footprint. It then reviews efforts to reduce the size of AI models and making them computationally more efficient which are the objectives of the scale down concept. Next and by building on top of scaling down, they focus on making AI systems decentralized, more adaptable, and collaborative which they refer to as scalding out.
Authors list future directions for scalding down and out, discuss challenges and opportunities, and provide alternative view points as well.

**Position:**

Yes

**Position In Title:**

Yes

**Related Work:**

3

**Strengths And Weaknesses:**

**Strengths**
- The paper does a decent job reviewing scaling up efforts and provides interesting background knowledge for the readers.
- The selected topic and proposed direction is relevant to many ongoing discussions within the AI community and can inspire further discussions in the ICML community.

**Weaknesses**
- The title of the paper suggests that the proposed framework of scaling down and out is relevant to all AI systems. However, based on the discussions in the paper, it looks like the proposed framework is mainly suggested for large language models.
- Alternative views section of the paper does not cover active areas of research to add more reasoning capabilities to LLMs or build stronger multi-modal generative AI to build images/videos.

**Support:**

3

---

> ### Author Rebuttal · Authors · 2025-04-01
>
> > **W1: The title of the paper suggests that the proposed framework of scaling down and out is relevant to all AI systems. However, based on the discussions in the paper, it looks like the proposed framework is mainly suggested for large language models.**
>
> Thanks for the constructive feedback. The proposed framework of Scaling Down and Scaling Out are intended to be applicable across various AI systems, not solely for large language models. Techniques discussed in Scaling Down such as pruning, quantization and knowledge distillation, are not only used in LLM but also effective in other architectures including plain transformer and CNNs. Similarly, methods highlighted within Scaling Out such as federated learning and conditional control are general approaches used across AI systems and are not specific to LLM scenarios. However, we agree that the current discussion and examples provided in our paper primarily emphasize large language models. This is due to that LLMs currently represent one of the most popular area within AI research, making them ideal for illustrating our framework. We will highlight the general applicability of the proposed framework in our revised paper.
>
> > **W2: Alternative views section of the paper does not cover active areas of research to add more reasoning capabilities to LLMs or build stronger multi-modal generative AI to build images/videos.**
>
> Thanks for the nice concern. In Alterview View Section, we presented an alternative viewpoint, which is "Scaling Up remains a viable trajectory despite the challenges". The discussion in this section are mainly among the potential ways of addressing the bottleneck of Scaling Up. We agree that this section could indeed benefit from discussing additional motivations behind Scaling Up, such as enhancing reasoning capabilities of LLMs and enabling more powerful multi-modal generative AI models.
>
> To address this, we plan to include the following additional discussion in our revised Alternative Views section:
>
> _Increasing the scale of AI models can achieve significant advancements in their reasoning and generalization capabilities, as evidenced by many foundaction models like LLaMa, Qwen and DeepSeek. Furthermore, Scaling Up has enabled transformative progress in multi-modal generative AI systems, enhancing the quality and fidelity of generated images and videos which smaller models often struggle to match. Models such as GPT-4o and Gemini demonstrate that larger scale can effectively integrate diverse modalities, leading to richer and more robust generative capabilities that have important real-world applications. Hence, despite the associated computational and economic challenges,Scaling Up is still meaningful as a critical strategy for pushing the boundaries of AI's reasoning and generative potential._
>
>
> > **Q1: What do authors think about architectural innovations and breakthroughs that can potentially increase the reasoning power of large language models? Before the introduction of attention and transformers, the field of deep learning was also plateauing and one could have argued that it's about time to focus on scaling down and out. While scaling down and out should happen naturally at every phase of innovation in the AI field, it should not be thought as a paradigm shift where training more complex models with more parameters or more advanced architecture is expected to show no additional gain and researchers should shift their attention to scaling down and out. It would be great to hear authors' thoughts on this.**
>
> Thank you for this insightful comment. Indeed, previous architectural innovations such as Transformer architecture have played critical roles in overcoming performance bottlenecks and significantly enhancing the reasoning capabilities of AI models.  Thus, we fully agree that future breakthroughs in model architectures such as novel attention mechanisms or new modeling paradigms could potentially lead to another revolution in Scaling Up.
>
> We appreciate the perspective that Scaling Down and Scaling Out tend to emerge naturally alongside each phase of progress in AI. In this paper, our position reflects where the field stands as of 2025, when the limitations of Scaling Up such as data saturation, computational constraints and diminishing returns from increasing model size are becoming more apparent. Given these challenges, we believe it is necessary to take Scaling Down and Out seriously, not as reactive steps, but as directions worth recognition and exploration.
>
> At the same time, we fully recognize that AI development is not static. Breakthroughs in architecture or other core technologies could very well shift the landscape again, just as attention mechanisms and Transformers did in the past. If that happens, the community will naturally re-evaluate its direction, including the one we propose here. Our goal is not to present a final answer, but to offer a clear position that makes sense in this moment.

---

> > ### Comment · Reviewer_hbVy · 2025-04-05
> >
> > I'd like to thank the authors for their explanations and sharing their perspectives on my questions. I changed my overall recommendation to "Accept".

---

### Official Review · Reviewer_Gk88 · 2025-03-09

**Significance:** 3
**Argument Clarity:** 3
**Rating:** 2
**Confidence:** 4

**Questions:**

NA

**Discussion Potential:**

1

**Paper Summary:**

This paper advocates the position that future AI developments will happen with smaller and specialized transformer models. Arguments are made on multiple aspects of efficiency and predictions are made on future trends.

**Position:**

Yes

**Position In Title:**

Yes

**Related Work:**

3

**Strengths And Weaknesses:**

The position in this paper is of high relevance and importance to the ICML community. The arguments on efficiency are sound.

The main weakness of this paper is in addressing alternative views. The position implies that scaling up LLMs is a dead end from now on and it is a necessary part of the reasoning to support the position. However, Section 7 is very weak in arguing against scaling up and in the end concludes that the community should do both scaling up and scaling down/out. This becomes a position that most people would agree with and is unlikely to inspire discussions.

**Support:**

3

---

> ### Author Rebuttal · Authors · 2025-04-01
>
> **W1: The main weakness of this paper is in addressing alternative views. The position implies that scaling up LLMs is a dead end from now on and it is a necessary part of the reasoning to support the position. However, Section 7 is very weak in arguing against scaling up and in the end concludes that the community should do both scaling up and scaling down/out. This becomes a position that most people would agree with and is unlikely to inspire discussions.**
>
> Thank you for the constructive feedback. We would like to clarify our intention with regard to the "Alternative Views" section. Our paper clearly asserts the position that Scaling Up faces significant bottlenecks and that future research should shift more attention toward Scaling Down and Scaling Out. However, in accordance with the ICML call for Position Papers, we are required to include a section that objectively presents viable **opposing perspectives**. The submission guideline explicitly states:
>
> `The paper must include an “Alternative Views” section in the main body of the paper (not an appendix) that describes and addresses one or more viable (not strawmen) positions that are opposed to the paper’s position.`
>
> Based on this, our goal in Section 7 is not to reinforce our position or argue further against Scaling Up, but to present an alternative viewpoint that is opposed to the position we stated in our paper. Specifically, we present one alternative view "Scaling Up remains a viable trajectory despite the challenges". Our intent is to present this alternative viewpoint fairly in line with the submission requirements, and to provide a basis for potential discussion.
>
> Throughout the rest of the paper, our position still asserts that the Scaling Up faces significant challenges including low data quality, rising computational demands and unsustainable energy comsuption. These challenges motivate our position that the field should shift toward efficient and distributed approaches, as outlined through the frameworks of Scaling Down and Scaling Out.

---

### Official Review · Reviewer_Ksnq · 2025-03-11

**Significance:** 3
**Argument Clarity:** 3
**Rating:** 4
**Confidence:** 3

**Questions:**

My questions are directly linked to the weaknesses I mentioned above.


1. Should LoRA be categorized under Scaling Down, given that while it reduces trainable parameters, it increases inference parameters by adding a low-rank adaptation matrix to the original foundation model?

2. In the Scaling Down section, why is dataset pruning not discussed as a critical strategy for selecting the most useful training data, and how could relevant works on dataset pruning enhance this discussion?

3. How can the impact of Scaling Down and Scaling Out be effectively evaluated, and what standardized metrics or benchmarks should be used to compare these approaches with existing AI scaling strategies?

**Discussion Potential:**

3

**Paper Summary:**

The paper introduces a holistic AI scaling framework with three paradigms: Scaling Up, Scaling Down , and Scaling Out. While Scaling Up drives progress, Scaling Down and Scaling Out address bottlenecks, sustainability, and accessibility.  While Scaling Up drives progress, Scaling Down and Scaling Out address bottlenecks, sustainability, and accessibility. The work explores applications in healthcare, manufacturing, and content creation, alongside challenges in resources, ethics, and other development.

## update after rebuttal

My concerns are solved. I improved my score.

**Position:**

Yes

**Position In Title:**

Yes

**Related Work:**

2

**Strengths And Weaknesses:**

Strengths:

1. This paper presents a unified approach to AI scaling by integrating Scaling Up, Scaling Down, and Scaling Out, addressing both technical and practical challenges.

2. The concept of Scaling Out fosters decentralized AI systems, enabling collaborative intelligence and long-term adaptability in rapidly evolving technological landscapes.

3. Very comprehensive literature reviews regarding the all three perspectives (e.g., Scaling Up, Scaling Down, and Scaling Out).


Weaknesses:

1. I am wondering of if LORA should be categorized into Scaling down as the parameters of the low rank adaptation matrixs are added into the original foundation models apart from their own parameters. Although the trainable parameters are less, the inference parameters become more.


2. In the scaling down section, dataset pruning should be mentioned, which aims at finding out the most useful data for training. Some related works are provided as follows,

[a] He, M., Yang, S., Huang, T., & Zhao, B. (2024). Large-scale dataset pruning with dynamic uncertainty. In Proceedings of the IEEE/CVF Conference on Computer Vision and Pattern Recognition (pp. 7713-7722).

[b]Mahmoud, A., Elhoushi, M., Abbas, A., Yang, Y., Ardalani, N., Leather, H., & Morcos, A. S. (2024). Sieve: Multimodal dataset pruning using image captioning models. In Proceedings of the IEEE/CVF Conference on Computer Vision and Pattern Recognition (pp. 22423-22432).

[c]Zhang, X., Du, J., Li, Y., Xie, W., & Zhou, J. T. (2024). Spanning training progress: Temporal dual-depth scoring (tdds) for enhanced dataset pruning. In Proceedings of the IEEE/CVF Conference on Computer Vision and Pattern Recognition (pp. 26223-26232).

[d]He, Y., Xiao, L., & Zhou, J. T. (2023). You only condense once: Two rules for pruning condensed datasets. Advances in Neural Information Processing Systems, 36, 39382-39394.


3. The paper does not describe how to evaluate the impact of Scaling Down and Scaling Out, making it difficult to compare with existing AI scaling strategies.

**Support:**

3

---

> ### Author Rebuttal · Authors · 2025-04-01
>
> > **Q1&W1: Should LoRA be categorized under Scaling Down, given that while it reduces trainable parameters, it increases inference parameters by adding a low-rank adaptation matrix to the original foundation model?**
>
> Thanks for the nice question. LoRA is primarily designed for parameter-efficient tuning, which means it reduces the number of trainable parameters during fine-tuning, thereby significantly lowering training costs and computational overhead. Even if it does not always reduce inference cost directly, reducing training and fine-tuning cost still aligns well with the goals of Scaling Down.
>
> Importantly, LoRA offers flexibility in how it is used during inference. Its parameters can either be merged into the original model weights, which is the more common approach, or kept separate as a low-rank update matrix (Delta_W). When merged, the total number of inference parameters remains unchanged, supporting the case that LoRA does not expand the deployed model's size. Even in scenarios where the low-rank matrices are not merged, the overhead is typically small compared to the full model, and the flexibility allows LoRA to be adapted to a wider range of deployment needs.
>
> > **Q2&W2: In the Scaling Down section, why is dataset pruning not discussed as a critical strategy for selecting the most useful training data, and how could relevant works on dataset pruning enhance this discussion?**
>
> Thank you for highlighting the importance of dataset pruning. We agree that dataset pruning aligns with the principles of Scaling Down by identifying and retaining the most informative subsets of data, thereby optimizing resource efficiency and performance of AI models. We will discuss and include the suggested works in the revised paper:
>
> _Recent efforts in dataset pruning have aimed to reduce training costs while preserving model performance, particularly on large-scale datasets. Traditional snapshot-based pruning methods estimate sample importance using static predictions from early or late training epochs, but often fail to generalize across architectures or adapt to the dynamic nature of learning. To address these issues, He et al. \[a\] propose **Dynamic Uncertainty (Dyn-Unc)**, which captures prediction variance over time using sliding windows, enabling better distinction between hard and mislabeled samples. Building on the need to incorporate temporal dynamics, the **Temporal Dual-Depth Scoring (TDDS)** \[c\] method introduces a two-tiered scoring system that tracks both the stability and contribution of samples throughout training. By doing so, TDDS avoids the oversmoothing effect of simple averaging strategies and offers stronger generalization across models and pruning rates. In the multimodal setting, **Sieve** \[b\] addresses noise in web-crawled image-text pairs by leveraging image captioning models trained on high-quality data and sentence transformers tuned for semantic similarity. This results in a lightweight yet effective pruning signal that outperforms previous methods on the DataComp benchmark. Complementing these pruning approaches, **YOCO** \[d\] bridges the gap between dataset condensation and pruning by introducing two rules, which are Logit-Based Prediction Error and Balanced Construction that enable flexible resizing of condensed datasets without repeating the condensation process, offering a practical solution for constrained training environments. Together, these works reflect a shift toward dynamic, scalable, and generalizable data selection methods across both unimodal and multimodal domains._
>
> > **Q3&W3: How can the impact of Scaling Down and Scaling Out be effectively evaluated, and what standardized metrics or benchmarks should be used to compare these approaches with existing AI scaling strategies?**
>
> Thanks for this insightful question. To effectively evaluate Scaling Down and Scaling Out, standard benchmarks should extend beyond traditional metrics like accuracy to include computational efficiency indicators such as FLOPs, latency and energy consumption. For Scaling Down, it is important to consider deployment efficiency such as how many instances of the model can be run on limited hardware, how much performance can be achieved per dollar spent, and how models perform under strict latency or power constraints. Metrics like cost-per-inference, performance-per-watt, or capability-per-dollar can provide a better understanding of their practical value.
>
> For Scaling Out, evaluation should include ecosystem-level metrics that capture how well specialized models and interfaces scale across diverse real-world tasks. These may include the number of downstream applications supported and the robustness of distributed deployments. Benchmarks can be extended to include multi-agent collaboration tasks, real-world federated learning scenarios, or platform-level indicators such as the growth and diversity of fine-tuned models in open model stores (e.g., Hugging Face, GPTs in OpenAI’s ecosystem).

---

### Official Review · Reviewer_H3M3 · 2025-03-15

**Significance:** 3
**Argument Clarity:** 3
**Rating:** 4
**Confidence:** 4

**Questions:**

1. Is there a clearer definition or more tangible definition for scaling out that the authors could suggest? Alternatively, is there a persuasive reason not to make the definition more clear/tangible?

2. Is Scaling Down and Out already happening just by the nature of putting the scaled-up models into practice in cost-effective, practical, targeted ways? If so, is there a need for a strong position to advocate for them, or are they just a natural progression of scaled-up models (and/or a natural progression of the current saturation of the scaling market)?

**Discussion Potential:**

3

**Paper Summary:**

EDIT: I am responding here to the authors' last rebuttal comment: This is an excellent addition. The authors have addressed my remaining concerns with their examples/explanations of scaling out, and I would love to see this last rebuttal comment expanded in the paper (or even in subsequent papers -- it is a powerful point and a refreshingly common-sensical take on scaling). I am revising my score upwards. Thank you once again for the constructive dialogue.

Original review:

This paper takes the position that while scaling up is soon going to hit some bottlenecks, progress can better be made by scaling down (isolating useful parts of a wasteful architecture/learned model/other training paradigm, and replicating only that) and scaling out (seeing AI models as part of a larger interconnected ecosystem, and scaling that ecosystem).

Note that while some may feel the position stated in the title should be stronger, I think it is clear as is and meets the criteria of the call (it could, of course, use some revision, but I don't think that it should be a major concern because it serves a useful function of keeping the paper's argument clear, concise, and relevant).

**Position:**

Yes

**Position In Title:**

Yes

**Related Work:**

3

**Strengths And Weaknesses:**

Strengths:

The main premise of the paper is clear, and the arguments are timely. The authors make a persuasive case for looking past the type of scaling-fervor that has dominated the last few years, and instead looking to "scale" in different ways. The technical information provided, largely in review format, is well chosen and technically sound. The organization is largely strong and helpful to the piece, although it could be more cohesive overall.

One of the most interesting perspectives unexpectedly comes in the Alternative Views section, where the authors present future Scaling Up as a necessarily interdisciplinary endeavor (requiring advancements in quantum computing and HPC, among other things). This, to my knowledge, is an unusual framing; usually, scaling up is seen as the engineering-first approach, while interdisciplinary perspectives are often lumped under the category of unnecessary/likely to fall victim to the bitter lesson of the power of more data and more scale. This is also a perspective that could foster much discussion among the ICML community, and is perhaps worth elaborating in the main text, not just the Alternative Views section (perhaps as a way to conjoin Scaling Up with Scaling Out, and thus reinterpret Scaling Up).


Weaknesses:

At times, the paper's position seems to reflect trends already existing or growing within the AI research community. The authors could head off this type of criticism by addressing that shift (or the misperception of that shift) head-on, explaining why their take is/is not indicative of a natural progression of ideas about Scaling Up no longer being enough.

Of the three, scaling out is the most nebulous as currently written, but also has the most potential. Currently, it encompasses everything from federated learning to task-specific fine-tuning. This level of breadth takes away from a strong positional argument in its favor, because it seems to simply bucket everything that is not a scaling-based approach into this category. Further refinement of this section would make the paper stronger.

A more disciplined focus on the position/message and less on the review paper aspect would also make the paper a stronger entry for this particular track.

**Support:**

2

---

> ### Author Rebuttal · Authors · 2025-04-01
>
> > **W1: At times, the paper's position seems to reflect trends already existing or growing within the AI research community. The authors could head off this type of criticism by addressing that shift (or the misperception of that shift) head-on, explaining why their take is/is not indicative of a natural progression of ideas about Scaling Up no longer being enough.**
>
> Thanks for your thoughtful comment. We agree that some ideas regarding the limitations of Scaling Up and the subsequent shift towards alternative approaches indeed reflect certain trends within the AI research community. However, the contribution of our paper goes beyond merely recognizing these trends. Specifically, our work systematically defines and organizes the concepts of Scaling Up, Scaling Down and Scaling Out into a coherent and structured framework.
>
> We clarify relationships and dependencies among these scaling strategies. For instance, how insights from Scaling Up inspire and guide effective Scaling Down and how Scaling Down further supports efficient and ecological Scaling Out. Additionally, while some techniques relevant to Scaling Down and Out such as pruning, quantization or federated learning have been individually explored, they are often treated as isolated solutions to practial constraints. In this position paper, we also show how these existing methods can form a structured scaling paradigm that prioritizes efficiency, adaptability and real-world integration.
>
> > **W2&Q1: Is there a clearer definition or more tangible definition for scaling out that the authors could suggest? Alternatively, is there a persuasive reason not to make the definition more clear/tangible?**
>
> Thanks for the nice question. In our paper, Scaling Out can refer to the deployment of lightweight, specialized models derived from large foundation models across distributed and task-specific environments. These models are interconnected through structured interfaces such as APIs or intelligent agents, creating an ecosystem where intelligence is decentralized and responsive to local contexts. To make the definition more tangible and clearer, we define the Scaling Out as: A paradigm that leverages efficient, task-specific AI models derived from large-scale foundation models, distributed across networks and interfacing through modular, interactive systems.
>
> > **W3: A more disciplined focus on the position/message and less on the review paper aspect would also make the paper a stronger entry for this particular track.**
>
> Thank you for the constructive feedback. We understand that a position paper should focus on advocating a clear perspective rather than presenting an extensive literature review. In our current paper, we included technical content and references in order to support a specific position, which is the shift from traditional Scaling Up toward Scaling Down and Scaling Out. Each section builds toward this argument, rather than merely categorizing existing methods. While some parts of the paper may resemble a literature review, our intent is not to summarize prior work, but to use these methods as evidence for a broader position. However, we agree with the reviewers' opinion that the position could be foregrounded more clearly and literature review part can be simplified. We will further address this concern in the revised paper.
>
> > **Q2: Is Scaling Down and Out already happening just by the nature of putting the scaled-up models into practice in cost-effective, practical, targeted ways? If so, is there a need for a strong position to advocate for them, or are they just a natural progression of scaled-up models (and/or a natural progression of the current saturation of the scaling market)?**
>
> Thanks for the thoughtful question. We agree that aspects of Scaling Down and Out are already beginning to emerge in practice, which is driven largely by the increasing cost, complexity and diminishing returns of Scaling Up. However, these approaches are still not becoming mainstream. Major new models such as Grok-3 from xAI or DeepSeek-V3 continue to focus on scaling up in size and compute, reinforcing the idea that “bigger is better” remains the dominant narrative in the field.
>
> Our motivation for writing this paper is because Scaling Down and Out are happening quietly, often as engineering practices, such as deployment optimization and edge adaptation, but are rarely treated as a strategic direction within the core of AI research. Despite real limitations in data, hardware and energy, there is still comparatively little public framing of Scaling Down or Out as equally valid, strategic alternatives to Scaling Up. By clearly formalizing and naming this shift, we hope to encourage the community to take Scaling Down and Out more seriously.

---

> > ### Comment · Reviewer_H3M3 · 2025-04-04
> >
> > Thank you to the authors for engaging with the review and for providing additional explanations/clarity, including regarding their motivations. I think the authors have done a good job of engaging with the major concerns. While the definition of scaling out is well taken, I think it would benefit from some revision or some lay-friendly examples (or even connections to non-network-related ideas of scaling out, including elements of human behavior and contextual detail). This needs further development, in my opinion, but that does not take away from the paper too much, and the motivation as explained in the rebuttal is sound. I will maintain my score. Thank you for the frank dialogue.

---

> > > ### Author Response · Authors · 2025-04-05
> > >
> > > Thank you for acknowledging our rebuttal and for your thoughtful feedback! We are pleased that the current definition of Scaling Out was well received, and we fully agree that supplementing it with more concrete examples will further enhance its clarity.
> > >
> > > The technical definition as stated in the rebuttal is ''Scaling Out as a paradigm that leverages efficient, task-specific AI models derived from large-scale foundation models, distributed across networks and interfacing through modular, interactive systems''. Building on that, we include clearer analogies and real-world illustrations to make the concept more tangible.
> > >
> > > 1. We first draw an analogy to **human society**. In a functioning society, individuals or groups often specialize in different roles like teachers, policemen, and engineers. Each adapts to their context while contributing to maintaining social order and driving progress. For example, a teacher adjusts their methods to meet the needs of students, while contributing to the broader educational infrastructure. This mirrors how Scaling Out in AI enables diverse, task-specific models to operate within their domains, while remaining connected through structured interfaces. Like a well-functioning ecosystem of social actors, these models respond to localized needs and contexts, enabling the overall system to remain adaptive, scalable, and resilient in real-world environments.
> > >
> > > 2. In modern **healthcare systems**, when diagnosing and treating a patient, hospitals do not rely on a single generalist. Instead, responsibilities are distributed across specialized professionals such as GPs handling basic diagnostics, radiologists interpreting medical images, surgeons performing targeted procedures and pharmacists managing prescriptions. Each operates based on their expertise and the specific contextual information they receive. These experts interact through structured protocols (e.g., referrals, shared records), and their collaboration results in an adaptive, efficient, and context-aware response tailored to individual needs. This reflects how AI systems under Scaling Out can coordinate modular intelligence across specialized sub-models.
> > >
> > > 3. In **workplace** settings, companies can redesign their organizational structures around AI-based agent hierarchies. Rather than treating AI as a standalone tool, these systems assign managerial roles to central AI agents that oversee teams of specialized sub-agents, each responsible for distinct tasks such as scheduling, writing documents, monitoring the market, or internal communication. Executives no longer delegate tasks manually to each sub-agent, instead, they assign high-level goals to the lead AI agent, which autonomously coordinates its team by distributing responsibilities according to task type and urgency. Employees interact with these agents through conversational interfaces like chatbots embedded in their workflow tools. This setup exemplifies Scaling Out by replacing monolithic systems with modular, coordinated agent-based ecosystems, where intelligence is distributed, contextualized, and embedded into the daily operation of human–AI hybrid teams.
> > >
> > > We thank the reviewer again for this helpful suggestion. We will include these analogies in the revised paper to improve the conceptual clarity of the Scaling Out paradigm.

---

### Decision · Program_Chairs · 2025-05-01

**Decision:**

Accept (poster)

**Comment:**

A well-written paper talking about scaling down and out instead of scaling up. Good survey of existing work and proposed solution is reasonable. The reviewers appreciated the timely topic and strong rebuttal. The reviewers all participated in discussion or acknowledged the rebuttal. Several reviewers have increased the ratings during this process.